# Evaluation of a Frozen Micro-Agar Plates of MAPt Antibiotic Susceptibility Test for Enhanced Bioterror Preparedness

**DOI:** 10.3390/antibiotics11050580

**Published:** 2022-04-26

**Authors:** Shahar Rotem, Ohad Shifman, Ronit Aloni-Grinstein

**Affiliations:** The Department of Biochemistry and Molecular Genetics, The Israel Institute for Biological Research, Ness-Ziona 74100, Israel; shaharr@iibr.gov.il (S.R.); ohads@iibr.gov.il (O.S.)

**Keywords:** antibiotic susceptibility test, bioterror, MAPt, *Bacillus anthracis*

## Abstract

There is an urgent need for rapid antibiotic susceptibility tests to improve clinical treatment and to support antibiotic stewardship, especially concerning the emergence of multi-drug-resistant bacteria. Nowadays this need is even more profound due to progress in synthetic biology procedures that may facilitate the malicious preparation of engineered antibiotic-resistant pathogens. We recently described a novel, rapid, simple, specific, and sensitive method named a Micro-Agar-PCR-test (MAPt) and showed its performance on clinical as well as environmental samples. The method does not require any isolation or purification steps and is applicable to a wide range of bacterial concentrations, thus allowing a short time to respond within a bioterror event (5–7 h for *B. anthracis*, 10–12 h for *Y. pestis*, and 16 h for *F. tularensis*). Ready-to-use reagents for this assay may add a level of preparedness. We examined the option of freezing pre-prepared MAPt agar plates and thawing them upon need. Our results show that adequate minimal inhibitory concentration (MIC) values are obtained with the use of thawed 6- and 12-month frozen agar plates. The ability to store MAPt micro-agar plates at −70 °C for a year, together with all other reagents required for MAPt, holds a great advantage for bioterror preparedness.

## 1. Introduction

Antibiotic resistance is a major concern in the clinic. Rapid antibiotic susceptibility tests (ASTs) may assure proper treatment and lifesaving in intensive care settings. This issue is more profound regarding bio-threat agents, mainly due to public anxiety and more importantly, in light of potential malicious intensions to engineer pathogens carrying antibiotic resistance. Preparedness means that will offer decision-making personnel prompt proper antibiotic treatment options are essential for prophylactic treatment of exposed individuals before symptoms appear and disease spread, and to prevent mortality among morbid individuals.

Recently we have published papers [1,2] describing a new method named MAPt (micro agar PCR test) that offers antibiotic susceptibility profiling simply and rapidly, both from clinical as well as from environmental samples, without the need for isolation, enrichment or purification steps. The method is based on the agar dilution assay combined with a specific and sensitive PCR detection step. The assay is performed in a 96-well agar plate and is applicable to a wide dynamic range of bacterial concentrations (~5 × 10^2^–5 × 10^9^ cfu/mL) [1,2]. The assay was applied to the three major Tier-1 bio-threat bacteria *Bacillus anthracis*, *Yersinia pestis* and *Francisella tularensis* against ciprofloxacin and doxycycline, the recommended first-line antibiotic treatment [3,4,5]. Adequate MIC values, both from clinical samples such as whole blood and blood cultures as well as from heterozygous bacterial contaminated environmental samples, were obtained in relatively short time frames (5–7 h for *B. anthracis*, 10–12 h for *Y. pestis*, and 16 h for *F. tularensis*), thus providing the ability to deliver antibiotic susceptibility profiling before the onset of sever morbidity [1,2].

A great benefit of an assay is the ability to store all essential components for long periods and to avoid the need to prepare fresh reagents. In the case of MAPt, all reagents are stored at −20 °C; nevertheless, the MAPt agar plates may be stored at 4 °C for only 2 months. The ability to freeze the plates and thaw them before use holds a great advantage. However, to the best of our knowledge, agar plates are known to be sensitive to freezing and, upon thawing, they lose their structure and firmness. One way to overcome this structure destruction is to apply pressure on the agar plate [6,7]; however, this is not applicable in a microbiology setting such as in antibiotic sensitivity testing where sterility is fundamental. We thought that in a 96-well format the surface tension of the micro plate will be enough to hold the agar integral structure inside the well. To that end, we froze at −70 °C MAPt plates targeted to detect ciprofloxacin and doxycycline MIC values and examined the MIC values obtained using thawed plates.

## 2. Results

We chose the fast-growing *B. anthracis* as the model bacteria to examine the option to use thawed frozen MAPt plates. Fast-growing bacteria are most challenging regarding rapid AST as the outcome of their infection without proper treatment can be fatal in no time. MAPt plates were prepared and stored as described in the Materials and Methods. The plastic bag sealed plates were frozen at −70 °C. As a preliminary experiment, we thawed plates at room temperature after 6 months and determined MIC values for *B. anthracis* to ciprofloxacin and doxycycline as described previously [1,2]. Adequate MIC values were obtained for both antibiotics on 6-month frozen plates (unshown data). Next, MAPt plates were thawed after 12 months of storage at −70 °C, and MIC values of *B. anthracis* to ciprofloxacin and doxycycline were determined and compared to the values obtained from fresh plates and to the standard microdilution test. The assay was performed on two bacterial concentrations, ~10^4^ cfu/mL and ~10^5^ cfu/mL, for each tested antibiotic. The MIC values obtained were 0.125 µg/mL for ciprofloxacin and 0.016 µg/mL for doxycycline at both tested bacteria concentrations (Table 1). Similar MIC values were obtained using fresh MAPt plates and by the standard CLSI microdilution test [8]. (See Appendix A). 

## 3. Discussion

The ultimate preparedness means for a bioterror event involving bacteria as a vector of disease would be the accessibility to a rapid AST that would provide an antibiotic susceptibility profile within a time frame of clinical relevance. MAPt may serve as a potential assay for such a scenario. To maximize the potential of MAPt we examined whether MAPt agar plates may be pre-prepared and kept frozen until use. This will allow preparation in advance of all components of the MAPt assay. Our results show that the Mueller–Hinton-MAPt agar plates can be frozen for at least one year. The same MIC values were obtained with the use of the thawed frozen plates compared to the values obtained with fresh plates. Both the agar and the antibiotics embedded in the agar did not change their performance regarding growth capacity and antibiotic activity compared to the freshly made MAPt plates.

## 4. Conclusions

Mueller–Hinton-MAPt agar plates can be frozen for at least a year, thus MAPt has the potential to serve as a ready-to-use kit for rapid antibiotic susceptibility profiling providing decision-making personnel with clinical options.

## 5. Materials and Methods

### 5.1. Bacteria Strain

*Bacillus anthracis* Vollum ∆pXO1 ∆pXO2 [9] was grown at 37 °C on BHI-A (BD 241830 Sparks, MD, USA). Colony forming units (cfu) counts were determined by platting 100 μL of serial tenfold dilutions in phosphate-buffered saline (PBS, Biological Industries, Beth Haemek, Israel).

### 5.2. Preparation of MAPt Agar Plates

MAPt agar plates were prepared with Mueller–Hinton agar (BD 225250, Sparks, MD, USA) according to the manufacturer’s guidelines. Agar dilution was conducted essentially as defined in CLSI standard M07 [8]. After autoclaving, the agar was cooled to 50 °C and 40 mL were aliquoted to 50 mL tubes to which the tested antibiotic ciprofloxacin (ciproxin 200, Bayer, Germany) and doxycycline (Sigma–Aldrich D9891, Rehovot, Israel) was added with an Eppendorf Multipette Dispenser. An amount of 10× of antimicrobial solution was diluted by two-fold serial dilutions in master tubes. Next, one part of the 10× antimicrobial solution was added to nine parts of molted agar. Antibiotics free agar served as growth control. An amount of 150 μL aliquots of the antibiotic-supplemented melted agar were divided into a 96-well plate. The MAPt agar plates were individually sealed in plastic bags and frozen at −70 °C. Plates were thawed at room-temperature when needed.

### 5.3. MAPt Assay

Tested samples (in duplicates) of 10 microliters were plated in different wells of the MAPt plates containing different concentrations of the tested antibiotics. The MAPt plates were incubated at 37 °C for 6 h. Then, the bacteria were obtained from the MAPt plate with 150 μL of PBS. 100 μL of the recovered bacteria was added to 100 μL of Triton buffer (20% Triton-X-100 in TE, Sigma–Aldrich, Rehovot, Israel) and the samples were heated for 30 min at 100 °C to sterilize the sample and extract the DNA. A sample of 5 μL was evaluated by qPCR using a 7500 Real-Time PCR system (Applied Biosystems, Bedford, MA, USA).

### 5.4. qPCR Reaction

The qPCR reactions were performed in 30 μL volumes containing 2.3 μL of 20 mg/mL bovine serum albumin (BSA; Sigma–Aldrich, A2153, Rehovot, Israel), 15.05 μL SensiFAST Probe Lo-ROX Mix (Bioline BIO84005, TN, USA), 3.05 μL forward primer (5 pmol/μL), 3.05 μL reverse primer (5 pmol/μL) and 1.55 μL TaqMan probe (5 pmol/μL), and 5 μL of DNA extract. The primers and probes used were [10,11]:PL3_F: AAAGCTACAAACTCTGAAATTTGTAAATTG.PL3_R: CAACGATGATTGGAGATAGAGTATTCTTT.Tqpro_PL3: FAM-AACAGTACGTTTCACTGGAGCAAAATCAA-BHQ-1.

The PCR thermal conditions were: 3 min at 60 °C followed by 40 cycles of 15 s at 95 °C and 35 s at 60 °C.

### 5.5. Quantification of Bacterial Growth Inhibition by qPCR

Bacterial quantification by qPCR was estimated using the Ct value, which was evaluated by the 7500 real-time PCR system Sequence Detection Software (version 1.4). The relative change in bacterial growth between an untreated control and antibiotic-treated sample (designated as FC) was calculated by the equation FC = 2^−ΔCt^ where ΔCt is the difference between the Ct of sampled bacteria compared to the Ct of the untreated control sample. A 10-fold variation between the antibiotic-treated and untreated samples is reflected by a ΔCt = 3.3 (log_2_10), in an efficient PCR. The MIC was determined as the lowest antibiotic concentration that reduced growth to ΔCt ≥ 3.3, which correlated with the lack of visible growth by the standard AST.

### 5.6. MIC Determination by Broth Microdilution

Standard broth microdilution was conducted in cation-adjusted Mueller–Hinton broth (CAMHB; BD 212322, MD, USA) according to the CLSI guidelines [8]. An inoculum of 5 × 10^5^–1 × 10^6^ cfu/mL suspended in CAMHB was added at a 1:1 volume to a 96-well plate (TPP, Cat# 92696) containing duplicates of two-fold serial dilutions of ciprofloxacin (ciproxin Teva, Israel) or doxycycline (Sigma–Aldrich D9891, Rehovot, Israel) in CAMHB at a final volume of 0.1 mL. Bacteria grown in CAMHB without the addition of the antibiotics served as growth controls in each assay. The 96-well plate was incubated at 37 °C for 20 h. MIC values were defined as the lowest antibiotic concentration that no growth was shown by visual inspection.

## 6. Patents

A patent application (IL270342) for the described antibiotic susceptibility test (MAPt) was filed by the Israel Institute for Biological Research.

## Figures and Tables

**Table 1 antibiotics-11-00580-t001:** Evaluation of MIC values for *B. anthracis* to ciprofloxacin and doxycycline after 12 months of freezing.

	MIC of Ciprofloxacin (µg/mL)			MIC of Doxycycline (µg/mL)		
Bacterial Conc. (cfu/mL)	12 Months ^1^	Fresh Plates ^2^	Standard CLSI Microdilution ^3^	12 Months ^1^	Fresh Plates ^2^	Standard CLSI Microdilution
10^4^	0.125	0.125	0.125	0.016	0.016	0.016
10^5^	0.125	0.125	0.125	0.016	0.016	0.016

^1^ N = 3 batches of 3 plates, 9 plates in total. ^2^ N = 3 plates. ^3^ N = 3 plates.

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
