# Peer review of "Evaluation of a Frozen Micro-Agar Plates of MAPt Antibiotic Susceptibility Test for Enhanced Bioterror Preparedness"

_antibiotics, 2022, doi:10.3390/antibiotics11050580_

Round 1
Reviewer 1 Report
Dear Authors,
in your brief report you describe a supplement to the method you have developed for the rapid testing of MIC by MAPt. The aim was to optimize a rapid test start of samples and to answer the question whether 1 year stored deep-frozen 96 well plates give the same result as fresh 96 well plates.
I am a big fan of method optimization and think your developed method is overall very fascinating and I see great potential in it. But I think a little improvement is still needed.
Introduction
Line 42-44: This part is somewhat incomprehensible if you have not read the previous paper in which these numbers were explained.
Line 49: This timepoints were not used for environmental samples. Please check and change.
Results
Please provide all data in a supplement, also the data of the plates that were thawed after 6 months (line 75) and different inoculum densities. Did you really have the same result for all plates and thus no standard deviation? This is surprising, especially since in the previous paper they gave an MIC of 0.063 µg/ml for ciprofloxacin instead of the current 0.125µg/ml.
Table 1: since both previous papers show data for ciprofloxacin first and then data for doxycycline, i would keep this order in this publication as well.
Discussion
Since only B. anthracis was tested on MH agar, the statement of using deep frozen 96 well plates can probably be transferred to Y. pestis. For the testing of F. tularensis, however, Cystine Heart Agar was used, if I understood correctly. Therefore, testing the stability of the antibiotics in this agar would also be necessary. So I would be careful with the generalization that all MAPt plates can be frozen. It would also be nice to have some kind of outlook by showing further developments, e.g.: do you need different primers for other B. anthracis strains? Is an application for very slow-growing bacteria such as mycobacteria conceivable? Are you planning to test other Tier-1 bio-threat bacteria?
Material and Methods
The material and methods section is mostly the same as in the previous papers, which is completely fine. Nevertheless, I would like to have some additions please.
Line 116: isn’t there a newer document than CLSI 2018?
Line 121: how is the agar transferred to the wells without losses due to gelation? Machine? special pipettes? ambient temperature?
Line 122: Storing in plastic bags and freezing is the core of this paper. Therefore, it would be important to have a very detailed description of the material and the processing (sealing process? size of the bags, how to freeze, how to store (how tight, what orientation).
Line 129: Are there samples tested or do you mean the different inoculum densities?
Line 131: is there a centrifugation step needed? If yes what time, temperature and g?
Line 136: What PCR machine did you use? Did you perform a melting curve and verified the product by agar gel electrophoresis?
Line 151: Very important: please check the formula! What is the range of you Ct values?
Line 168: I do not fully understand the method: Why is every hour measured? Are growth curves important? What is the range of OD? Is the background well delineated? What are the criteria for growth after 20h when no visible growth can be observed? It would be good to put these data and curves in a supplement.
Has a quality control strain been tested at the same time or is there CLSI data for this strain?
Nevertheless, I like your work very much and wish you success with this publication and all the following ones.
Reviewer 2 Report
The present manuscript entitled "Evaluation of frozen micro-agar plates of MAPt antibiotic susceptibility test for enhanced bioterror preparedness" by Shahar Rotem, Ohad Shifman, and Ronit Aloni-Grinstein (antibiotics-1691834) describes an examination of the option of freezing pre-prepared micro-agar plates (MAPt) and thawing them upon need in terms of antibiotic susceptibility test. The adequate minimum inhibitory concentration (MIC) values were received with the use of thawed 6 and 12 months frozen agar plates. The possibility to store MAPt microagar plates for a year with all necessary reagents is an excellent advantage for bioterror preparedness.
The present article – a brief report – is written correctly and well-structured. The paper meets Antibiotics' requirements, and I recommend the article for publication in Antibiotics following the common editing stage. My current decision is a minor revision. More specific comments and observations are presented below.
- Abstract. When you mention a short time, please add the value in parentheses.
- Please add an explanation to MIC.
- Page 2, line 50. “providing the ability to provide”, please rephrase it.
- Page 2, line 63. “examined” should be instead of “exaimined”.
- Page 3, line 108. “Levy et al., 2014” should be mentioned in references and written as a number.
- Please add the conclusions section.
- Page 4, line 116. “(CLSI, 2018)” should be written as a number.
- When listing reagents and instruments, in addition to companies, the countries of origin should also be added.
- References should be recorded in accordance with the journal's requirements.
- Please check the formatting (font, size) throughout the manuscript and adjust it to your requirements.
I hope that the comments presented will help improve the article.
